# Association between Global Monkeypox Cases and Meteorological Factors

**DOI:** 10.3390/ijerph192315638

**Published:** 2022-11-24

**Authors:** Md. Aminul Islam, Sarawut Sangkham, Ananda Tiwari, Meysam Vadiati, Mohammad Nayeem Hasan, Syed Toukir Ahmed Noor, Jubayer Mumin, Prosun Bhattacharya, Samendra P. Sherchan

**Affiliations:** 1Advanced Molecular Lab, Department of Microbiology, President Abdul Hamid Medical College, Karimganj 2310, Bangladesh; 2COVID-19 Diagnostic Lab, Department of Microbiology, Noakhali Science and Technology University, Noakhali 3814, Bangladesh; 3Department of Environmental Health, School of Public Health, University of Phayao, Muang District, Phayao 56000, Thailand; 4Department of Food Hygiene and Environmental Health, Faculty of Veterinary Medicine, University of Helsinki, 00014 Helsinki, Finland; 5Department of Health Security, Expert Microbiology Research Unit, Finnish Institute for Health and Welfare, 70701 Kuopio, Finland; 6Hubert H. Humphrey Fellowship Program, Global Affairs, University of California, Davis, 10 College Park, Davis, CA 95616, USA; 7Department of Statistics, Shahjalal University of Science & Technology, Sylhet 3114, Bangladesh; 8Joint Rohingya Response Program, Food for the Hungry, Cox’s Bazar 4700, Bangladesh; 9Platform of Medical and Dental Society, Dhaka 1214, Bangladesh; 10COVID-19 Research, Department of Sustainable Development, Environmental Science and Engineering, KTH Royal Institute of Technology, Teknikringen 10B, SE 10044 Stockholm, Sweden; 11Department of Biology, Morgan State University, Baltimore, MD 11428, USA; 12Department of Environmental Health Sciences, School of Public Health and Tropical Medicine, Tulane University, New Orleans, LA 70118, USA

**Keywords:** Monkeypox disease (MPXD), monkeypox virus (MPXV), monkeypox (MPX), daily cases, zoonotic disease, meteorological factors, mathematical models, temperature and dew point, treatment and vaccine

## Abstract

The emergence of an outbreak of Monkeypox disease (MPXD) is caused by a contagious zoonotic Monkeypox virus (MPXV) that has spread globally. Yet, there is no study investigating the effect of climatic changes on MPXV transmission. Thus, studies on the changing epidemiology, evolving nature of the virus, and ecological niche are highly paramount. Determination of the role of potential meteorological drivers including temperature, precipitation, relative humidity, dew point, wind speed, and surface pressure is beneficial to understand the MPXD outbreak. This study examines the changes in MPXV cases over time while assessing the meteorological characteristics that could impact these disparities from the onset of the global outbreak. To conduct this data-based research, several well-accepted statistical techniques including Simple Exponential Smoothing (SES), Auto-Regressive Integrated Moving Average (ARIMA), Automatic forecasting time-series model (Prophet), and Autoregressive Integrated Moving Average with Explanatory Variables (ARIMAX) were applied to delineate the correlation of the meteorological factors on global daily Monkeypox cases. Data on MPXV cases including affected countries spanning from 6 May 2022, to 9 November 2022, from global databases and meteorological data were used to evaluate the developed models. According to the ARIMAX model, the results showed that temperature, relative humidity, and surface pressure have a positive impact [(51.56, 95% confidence interval (CI): −274.55 to 377.68), (17.32, 95% CI: −83.71 to 118.35) and (23.42, 95% CI: −9.90 to 56.75), respectively] on MPXV cases. In addition, dew/frost point, precipitation, and wind speed show a significant negative impact on MPXD cases. The Prophet model showed a significant correlation with rising MPXD cases, although the trend predicts peak values while the overall trend increases. This underscores the importance of immediate and appropriate preventive measures (timely preparedness and proactive control strategies) with utmost priority against MPXD including awareness-raising programs, the discovery, and formulation of effective vaccine candidate(s), prophylaxis and therapeutic regimes, and management strategies.

## 1. Introduction

Viral zoonotic pathogens are of significant concern for global public health. Although the COVID-19 public health emergency status on cessation, the world is challenged by another new reemerging outbreak of the Monkeypox disease (MPXD). Although there is extensive literature on the ongoing pandemic (COVID-19) caused by the zoonotic SARS-CoV-2 virus [1,2,3,4], very little is known about the Monkeypox (MPX) outbreak––which has already infected 65,353 people, with 20 death cases in 111 countries (as of 9 November 2022). The top three countries with the highest MPXD cases include the US (26,385), Brazil (7300), and Spain (7083) (WHO, 2022) [5]. The World Health Organization (WHO) declared the ongoing monkeypox outbreak a Public Health Emergency of International Concern, whereas the US declared the growing monkeypox outbreak a national health emergency [6].

Monkeypox virus is a zoonotic virus, which is encapsulated by double-stranded (ds) DNA and belongs to the Orthopoxvirus genus under the Poxviridae family with other notable viruses [i.e., cowpox, Vaccinia virus, and variola (smallpox)] [7,8,9]. The MPXD spreads by direct contact with infected wild animals or, human-to-human transmission is conceivable [8]. The common clinical symptoms of MPX patients are similar to other *pox viruses*, such as Lymphadenopathy, chills, back pain, and headache (Figure 1) [10]. The reservoir animal species of the MPXV are still unclear despite increasing evidence of acute or previous infection in various animals. Mice (*Mus musculus*), Gambian pouched rats (*Oryctolagus cuniculus*), ferrets, woodchucks (*Marmotamonax* sp.), jerboas (*Jaculus* sp.), and raccoons (*Atherurus africanus*) have all been reported to be infected with MPXV [11,12,13]. The chain of transmission includes terrestrial rodents to arboreal rodents, arboreal rodents to non-human primates, and non-human primates to terrestrial rodents or vice versa. Transmission from terrestrial rodents, arboreal rodents, and non-human primates to humans or vice versa was also found [3,14].

Although the smallpox disease has been eradicated, many questions are still unclear. One of the most important ecological questions is the correlation of case fatality rates with metrological parameters, as it is thought to be a winter disease in industrialized countries. One research article with time series data from historical databases reported that dry climatic conditions favor smallpox virus transmission [15].

Nonetheless, despite the significance of MPXV predictions worldwide, there is a lack of comparative studies that examine how MPXV cases change over time while assessing meteorological characteristics. To fill this research gap, we compare the capability of four promising models, including Simple Exponential Smoothing (SES), Auto-Regressive Integrated Moving Average (ARIMA), Automatic forecasting time-series model (Prophet), and Autoregressive Integrated Moving Average with Explanatory Variables (ARIMAX) to delineate the correlation of meteorological factors on daily Monkeypox cases worldwide. The precision and accuracy of the models are compared based on error criteria indices while a comprehensive database from different continents issued by global organizations is used to evaluate the proposed models. A globally concerted effort toward the practice of open data sharing and open science priorities is required to generate data and gain adequate knowledge on Monkeypox, which is crucial to rapidly tackle the epidemic and reduce long-term human MPXV infection [16,17,18,19,20,21].

## 2. Materials and Methods

### 2.1. Daily Confirmed Monkeypox Cases

We derived data on confirmed global Monkeypox cases of infected countries (daily new confirmed cases per million population) from global data (https://ourworldindata.org/monkeypox; accessed on 10 October 2022), (Appendix A). The data were collected from 6 May 2020 to 10 October 2022, from available online servers.

### 2.2. Meteorological Factors

We used NASA’s Prediction of Worldwide Energy Resources webpage [22] where capital cities are counted for this study, on a daily interval to collate meteorological variables including rainfall (mm), relative humidity [RH, (%)], temperatures (°C), surface pressure (kPa), dew point (°C), and wind velocity (m/s) at 10 m height (Maximum Wind Speed) available from https://power.larc.nasa.gov/data-access-viewer (accessed on 10 October 2022) (Appendix A).

### 2.3. Time Series Models

The sampled variables were used to select four promising time series models namely SES, ARIMA, ARIMAX, and Prophet, which were employed to examine the trend in MPX cases. We forecasted the number of new MPXD cases for 30 subsequent days while using the SES model as a benchmark to assess the prediction accuracy of other models.

#### 2.3.1. SES Model (Simple Exponential Smoothing)

The SES model is the most used model for analyzing time series data sets [23]. The SES is an easy-to-use tool that treats data as varying around a constant mean [24]. It is a univariate time-series forecasting approach in which there is no evidence of trend or seasonality in the data. Based on the most recent data, which are given more weight, a weighted mean is used to predict future values, and less importance is given to older observations. The SES model in this study is performed using the R package “fpp2” [25].

The equation of the SES model can be expressed as:(1)$$F_{t}=F_{t-1}+\alpha \left(A_{t-1}-F_{t-1}\right)$$
where $\;A_{t}$ is the actual value of the series at time t, $F_{t}$ is the forecast value of the series at time t, and α is a weighting parameter that takes a value between 0 and 1.

#### 2.3.2. ARIMA Model (Auto-Regressive Integrated Moving Average)

We applied the ARIMA, a mathematical, data-driven methodology that forecasts the pattern of daily monkeypox incidence by utilizing the structure of the data itself. To produce stationary time series, the general linear stochastic model incorporates autoregressive, moving-average models, and differencing factors [26]. In a conventional autoregressive model, the future values of variables of interest are predicted using a linear combination of the past values. Similar to a regression model, the moving-average model incorporates the errors from past projections. The ARIMA produces accurate results in the absence of seasonality in the data [27]. The R package “forecast” was utilized in this study to run the ARIMA model expressed as [23]:(2)$$y_{t}=c+\varphi_{1}y_{t-1\;}+\ldots +\varphi_{p}y_{t-p}+\theta_{1}e_{t-1\;}+\ldots +\theta_{q}e_{t-q}+e_{t}$$
where $y_{t}$ is the difference at degree d of the original series of time series, $\varphi_{1}-\varphi_{p}$ are autoregressive model parameters, $\theta_{1}-\theta_{q}$ represents moving-average model parameters, and $e_{t}$ is white noise.

#### 2.3.3. Prophet Model (Automatic Forecasting Time-Series Model)

The Prophet model fits the data set relatively quickly and is utilized for observations with irregularities, a model for additive regression. The Prophet model’s components are a logistic gain curve trend, which selects the data’s growing points to identify variances in trends. The Fourier series can be used to simulate an annual seasonal component whereas dummy variables can represent a seasonal weekly element. The model functions effectively with historical data from various seasons and time series with significant seasonal fluctuation while accounting for missing data and outliers. As a result, time series generally benefit from this without losing data values or transferring in a trend or outlier. The R package “Prophet” is used to analyze time-series data using the Prophet model to track cases and fatalities of MPXV expressed as [28]:(3)$$y_{t}=g_{t}+s_{t}+h_{t}+\in_{t}$$

Equation (3) is used for Prophet analysis where $g_{t}$, $s_{t}$, and $h_{t}$ are model factors and $\in_{t}$ is used for non-periodic changes.

#### 2.3.4. Autoregressive Integrated Moving Average with Explanatory Variables (ARIMAX)

The ARIMA model accepts a direct relationship between the time-series values attempts to exploit these straight conditions in perceptions and arranges to extricate nearby designs, removing high-frequency commotion. In this model, the explanatory information variable (X) is added, which is called ARIMAX (*p*, *d*, *q*), for accurate interpretation [26].

#### 2.3.5. Empirical Evaluation

The prediction of case fatality rate prediction was evaluated using four time-series models which were compared with the benchmarks. The benchmark is permitted to gauge its competitors’ impact [29]. The SES model, which allows for errors or trend elements, is the most appropriate non-seasonal model for a time series analysis [30]. The execution of the time series models is examined and contrasted in this study to ensure the robust prediction, coefficient of determination (R^2^), root mean square error (RMSE), and mean absolute error (MAE).

#### 2.3.6. Statistical Analysis

Spearman rank correlation coefficients were used to study meteorological variables and confirm global daily Monkeypox cases. The evolution of MPX cases was examined using a time series model. We used the ARIMAX and negative binomial regression models to determine whether there is a link between MPX cases and deaths with meteorological variables using R software. The number of individuals with MPX infection is the dependent variable, whereas meteorological variables are the independent variables in both models. The negative binomial regression (NBR) model was employed to further examine variations in MPX infections among nations. The Poisson-gamma mixed distribution is the foundation of a negative binomial regression model, which is helpful in forecasting count-based data. The adoption of this technique is due to the non-negative integer values of the reported monkeypox cases (the number of monkeypox infections) and the higher variance of the dependent variable compared to its mean. The association is validated using the coefficient, 95% confidence interval (CI), and corresponding *p*-value.

## 3. Results

According to meteorological factors in Table 1, the mean for monkeypox cases is 420.02, with a standard deviation (SD) of 459.30, a minimum of 0, and a maximum of 1985 (Table 1). The global descriptive statistics for meteorological variables show the average daily Monkeypox confirmed cases between June to November 2022 is 420. The highest average daily confirmed cases (number of persons) by continent for North America is 5.12, followed by Europe (4.56), South America (1.92), Africa (0.25), Oceania (0.18), and Asia (0.15). Average daily temperature, dew/frost point temperature, relative humidity, precipitation, surface pressure, and wind speed are 21.21 °C, 14.49 °C, 70.25%, 4.53 mm/day, 96.32 kPa, and 2.11 m/s, respectively.

The empirical meteorological data from six continents in Appendix A shows that the average and standard deviation of temperature (°C) in Africa, Asia, Europe, North America, Oceania, and South America are 23.65 ± 5.61, 26.45 ± 6.73, 16.88 ± 5.99, 23.70 ± 4.65, 14.33 ± 4.01, and 19.46 ± 8.71, whereas dew/frost point temperatures (°C) are 16.61 ± 7.51, 14.75 ± 7.41, 10.80 ± 4.31, 23.88 ± 4.78, 15.77 ± 7.15, and 14.81 ± 9.32, respectively. We observe high relative humidity in Oceania (80.48%), followed by South America (78.19%), North America (77.89%), Europe (70.54%), Africa (70.33%), and Asia (58.29%). The average mean of precipitation including rain, snow, drizzle, graupel, ice pellets, hail, and sleet (mm/day) in North America, South America, Africa, Asia, Oceania, and Europe are 9.14, 6.59, 4.54, 3.96, 3.02, and 2.44, respectively. In this study, the observation period shows that the surface pressure (kPa) for Africa, Asia, Europe, North America, Oceania, and South America are 94.28, 95.35, 96.77, 96.21, 100.63, and 96.41, and the average daily wind speed (m/s) was 1.75, 2.49, 1.88, 2.88, 5.05, and 0.86, respectively. The highest standard deviation in Appendix A was recorded in relative humidity in Africa (17.66), Asia (23.29), Europe (14.61), North America (14.19), Oceania (8.29), and South America (14.76). In contrast, wind speed has the lowest standard deviation in Africa (0.85), Asia (1.29), Europe (1.33), North America (2.10), Oceania (1.81), and South America (0.91). The highest standard deviation for monkeypox cases occurs in North America (33.21 per person), whereas the lowest occurs in Asia (0.83 per person).

The Spearman rank correlation coefficients among meteorological variables and confirmed daily cases of MPX suggest a significant but weak correlation between MPX cases and meteorological variables (Figure 2). The mean temperature and dew/frost point temperature exhibit a significant positive but weak correlation with daily MPX daily cases (*r* = 0.56, *p* < 0.05 and *r* = 0.48, *p* < 0.05, respectively). However, mean relative humidity and wind speed has a negative association with daily Monkeypox cases (*r* = −0.36, *p* > 0.05 and *r* = −0.25, *p* > 0.05, respectively). The global distribution of daily MPX cases is reported mostly in North America, Europe, and South America (Figure 3) (Appendix A). About 78,924 cumulative confirmed cases have been documented globally as of 9 November 2022, from 6 May 2022. The top continents include North America (33,397), followed by Europe (25,037), South America (18,213), Africa (935), Asia (368), and Oceania (173).

We observed a constant trend between observed and predictive global MPX cases in the SES model, with R^2^, RMSE, and MAE of 33.20%, 374.37, and 260.44, respectively (Table 2 and Figure 3). The prediction results from the time series models in Figure 4 show confirmed and predicted Monkeypox cases from June to November 2022. We observed a robust, increasing trend between observed and predictive global MPX cases with R^2^, RMSE, and MAE of 63.04%, 276.04, and 182.39 for the ARIMA model but64.88%, 271.45, and 185.04 for the ARIMAX model (Table 2). The Prophet model performed better in terms of accuracy (i.e., the coefficient of determination is more significant along with lower rates) compared to other models (i.e., R^2^ = 56.47%, RMSE = 378.68, and MAE = 298.32). According to the forecast in all models except SES, the number of monkeypox cases is expected to increase considerably in ten subsequent days (Figure 3). In the M–K trend analysis, we identified an increasing trend of daily monkeypox cases (*p <* 0.001 and tau = 0.238). Using Sen’s slope test, we find that the slope is 1.62 (95% CI: 0.84–2.76) (Table 2). In the ARIMAX and NB model, temperature (51.56 [95% CI: −274.55 to377.68] and 0.86 [−0.42 to 2.13], respectively) and relative humidity (17.32 [−83.71 to 118.35] and −0.14 [−0.53, 0.26], respectively) have an insignificant positive and negative association with monkeypox cases, respectively. However, precipitation is negatively associated with monkeypox cases (−19.41 [−57.25 to 18.42] and −0.05 [−0.19 to 0.08], respectively) in the ARIMAX model whereas dew point are negatively associated with monkeypox cases (−69.59 [−366.61 to227.43] and −0.61 [−1.93 to 0.71], respectively) in the ARIMAX and NB model (Table 3).

## 4. Discussion

The present data-based study examines the global association between meteorological factors and Monkeypox cases. Various vector-borne zoonotic viral diseases including COVID-19, dengue, Chikungunia, malaria, Ebola, and Zika are increasing with changes in climatic conditions [31,32,33,34,35]. Previous studies have shown that meteorological factors affect the growth and activity of respiratory viral diseases including SARS-CoV [36,37,38]. Experimental studies have shown that SARS-CoV-2 is highly active in conditions with low relative humidity and temperature, whereas genetic materials decay rapidly in an environment with high relative humidity and temperature [4,39,40]. When the virus is exposed to increased relative humidity, temperatures, and simulated solar light, the virus becomes even less stable (half-life, 3 min) [41,42]. In addition to the temperature factor, rainfall, and wind speed among other weather/meteorological factors influence the spread of COVID-19 [43,44].

Research on the relationship between meteorological characteristics and infectious diseases (such as avian influenza A/H5N1, SARS-CoV, and MERS-CoV) reported the significance of meteorological factors on the transmission of epidemics/pandemics [38]. For example, Sarkodie and Owusu, (2021) found a negative association between metrological parameters (e.g., wind speed, solar radiation, and humidity) and the COVID-19 pandemic in Iran. Sarkodie and Owusu, 2021, and Shi et al., 2020, reported that meteorological parameters are correlated with SARS-CoV-2 infection in China. Few studies reported similar results in different countries such as Malaysia (Suhaimi et al., 2020), the US (Sarkodie and Owusu, 2020), Norway [45], and Indonesia [38]. Monkeypox, a zoonotic disease (ZD) caused by an orthopoxvirus resulting from a smallpox-like disease in humans (Bunge et al., 2022), has become a public health concern worldwide. Since early May 2022, more than 40,000MPXV infections have been reported in more than a hundred countries across five regions, prompting WHO to declare Monkeypox on 23 June 2022, as an evolving threat and of moderate public health concern [46,47]. A study showed the daily Monkeypox cases to be mainly in North America, Europe, and South America [48], specifically on multicounty levels in the UK, Spain, Portugal, and the USA [49,50,51].

This study is the first to report the correlation between meteorological factors and daily Monkeypox cases across different countries. The study reveals the association between meteorological factors and the number of daily confirmed cases worldwide. The results show meteorological variables such as temperature are positively correlated with daily monkeypox cases, whereas wind speed has a negative correlation. Previous research findings reported that dry weather is favorable for influenza [52,53]. Low (dry) relative humidity in the range of 20 to 30% stimulates the spread of the influenza virus faster than relative humidity in higher percentages. A humidity of ≥80% is reported to slow down the spread of the flu [54]. Similarly, this study found that RH above 80% in the Oceania region has an average daily confirmed case of 0.28.

However, correlation studies reporting on macro-scale data such as country, community, or local level do not indicate association at the micro-scale. In addition, 95% of monkeypox transmissions are suspected to have occurred through sexual activities, hence, sharing activities with the patient such as touching, breathing, and eating together, may facilitate future spread by increasing the risk of infection [46]. Therefore, the route of exposure differs from SARS-CoV-2, which is primarily transmitted through aerosol droplet nuclides [55]. Other individual country-based studies are needed to determine the relationship between meteorological factors and daily monkeypox cases in countries with reported high prevalence. This study shows a statistically significant correlation between positive and negative daily meteorological factors and daily Monkeypox cases. This study failed to include interventions such as the movement of infected people from one place to another; for example, information on the infected persons and asymptotic or who are at risk of exposure. Additionally, climate factors, including climate change influence changes like the virus, thus, increasing the cross-species viral transmission risk [34,56,57,58]. Our prediction techniques such as Prophet, ARIMA, and SES show a predictive power of 59.40%, 51.78%, and 31.09%, respectively. Thus, the data obtained from this study is crucial in surveillance and response planning and public health prevention policies of diseases related to environmental factors that have the potential to spread to humans at the local level, nationally, and globally.

## 5. Limitation

As the infection is self-limiting, significant numbers of infected individuals may not seek clinical testing, and a large magnitude of cases can be unreported. The publicly available data may contain underreported statistics of Monkeypox-positive patients, which may affect our investigations. The original scenario of meteorological effects and vaccination may differ due to variability in air pollution, human immunity, individual migrations, mobility, behavior, habits, economic and lifestyle, and cultural conditions––which may affect the incidence of Monkeypoxby acting as confounders. In addition, our study was based on outdoor meteorological data; however, Monkeypox virus transmission can be affected quite differently by indoor air conditions. In future studies, these criteria could be included while evaluating the combined meteorological indicators and Monkeypox. As different season-based and long-time data are not available, in this study, seven months of monkeypox cases are used. Furthermore, this study has another drawback—sorting climate factors, as capital cities are counted in this study rather than all cities in a country, leads to a lack of abundant, available online, and valid data sets.

## 6. Conclusions

This study is the first to analyze the possible association between meteorological factors and daily MPX cases using current data, including all the infected countries. Monkeypox cases have a substantial association with meteorological factors such as temperature, dew/frost point, precipitation, relative humidity, and wind speed. Further investigations are required to understand the daily patterns of Monkeypox and the transmission of the virus at the local and community level. The results revealed that temperature and dew point can influence MPXV cases (51.56, 95% CI: −274.55,377.68) and (−69.59, 95%CI: −366.61,−227.43), respectively. The analysis of our results demonstrates that Monkeypox has no full vaccination measures yet; however, many countries are closely using the surveillance and monitoring measures of other countries. Our findings can be helpful for better understanding, monitoring, and controlling the transmission of MPXV.

## Figures and Tables

**Figure 1 ijerph-19-15638-f001:**
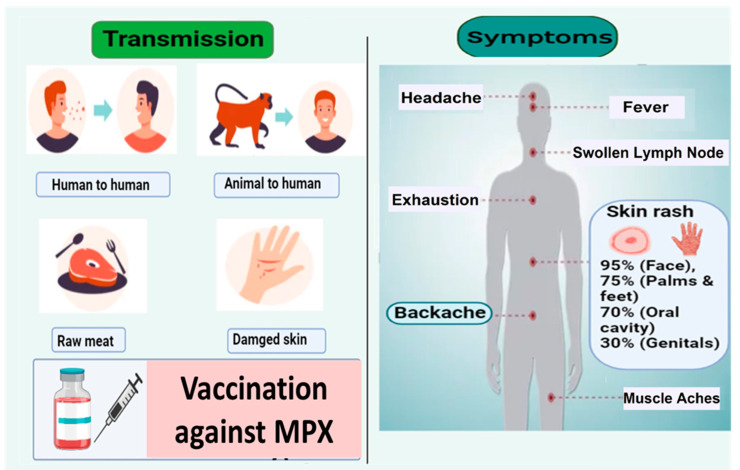
Symptoms, transmission, and treatment of MPXV. Common symptoms include chills, fever, swollen lymph nodes, muscle pain, tiredness, back pain, and rashes. Infected wild animals, raw meat, bites, scratches, and sex without protection, especially men with men, are known as the primary transmission methods, whereas secondary transmission from contaminated substances also occurs.

**Figure 2 ijerph-19-15638-f002:**
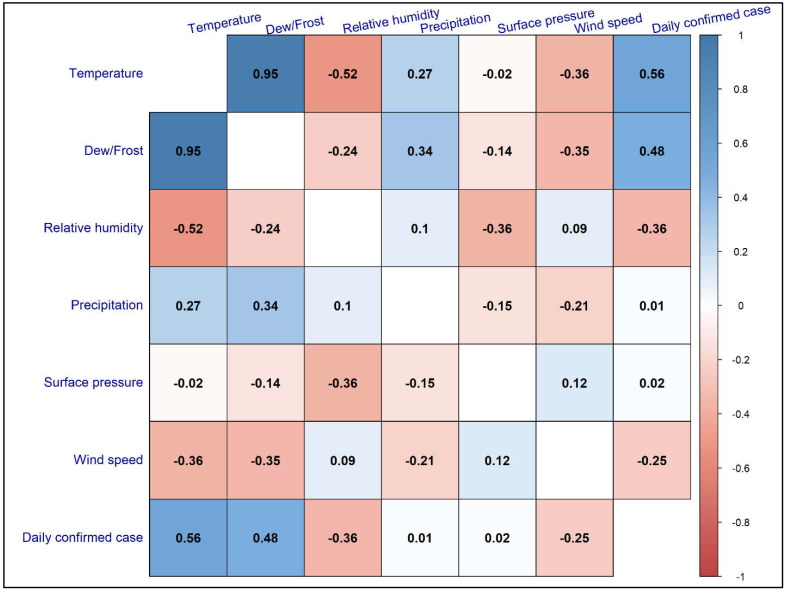
Spearman correlation rank coefficients between meteorological variables and confirmed global daily Monkeypox cases.

**Figure 3 ijerph-19-15638-f003:**
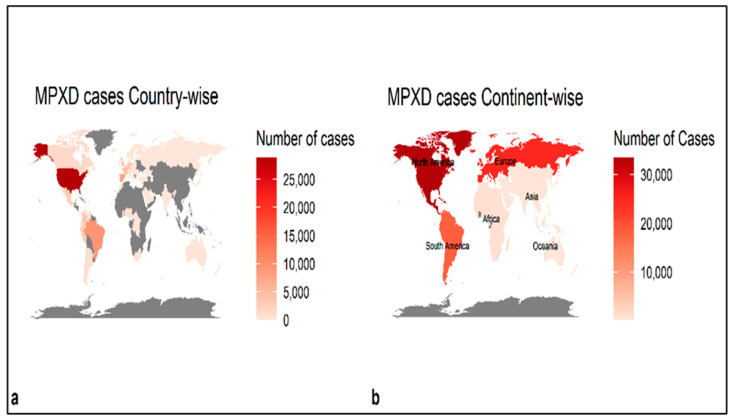
The global distribution of the total (cumulative) monkeypox cases in different countries and continents (10 November 2022).

**Figure 4 ijerph-19-15638-f004:**
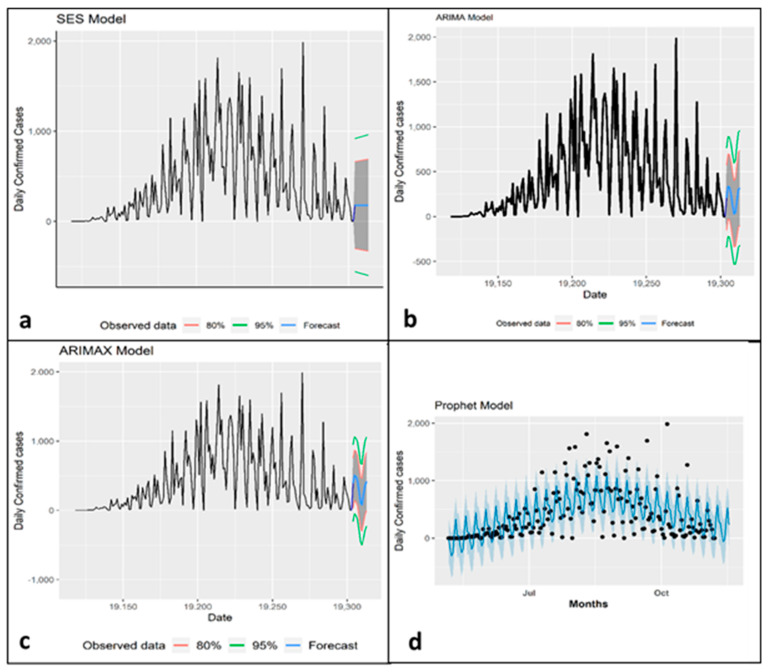
Time series plot of confirmed cases for different methods: (**a**) Simple Exponential Smoothing (SES), (**b**) Auto-Regressive Integrated Moving Average (ARIMA), (**c**) Autoregressive Integrated Moving Average with Explanatory Variable (ARIMAX), and (**d**) Automatic Forecasting time-series model (Prophet Model).

**Table 1 ijerph-19-15638-t001:** The global descriptive statistics for the meteorological variables.

Variables	Mean ± SD	Minimum	Maximum
Temperature (°C)	21.21 ± 1.38	17.45	23.24
Dew/frost point temperature (°C)	14.49 ± 1.17	11.34	15.93
Relative humidity (%)	70.25 ± 1.48	65.55	74.08
Precipitation (mm/day)	4.53 ± 1.44	1.87	9.54
Surface pressure (kPa)	96.32 ± 0.11	96.04	96.61
Wind speed (m/s)	2.11 ± 0.14	1.8	2.55
Daily confirmed cases (number of people)	419.57 ± 459.29	0	1977

**Table 2 ijerph-19-15638-t002:** The summary of SES, ARIMA, ARIMAX, Prophet, M–K trend, and Sen’s slope analysis.

Method and Period	R^2^	RMSE		MAE
**Simple Exponential Smoothing**				
Overall	33.20%	374.37		260.44
**Auto-Regressive Integrated Moving Average**				
Overall ARIMA	63.04%	276.04		182.39
**Auto-Regressive Integrated Moving Average with explanatory variables**				
Overall ARIMAX	64.88%	271.45		185.04
**Automatic Forecasting time-series model (Prophet Model)**				
Overall	56.47%	378.68		298.32
**Mann-Kendell trend analysis**				
	Tau		** *p* ** **-value**	
	0.238		<0.001	
**Sen’s slop test**				
	**Sen’s Slope**		**95% CI**	
	1.62		0.84 to 2.76	

RMSE: Root Mean Square Error; MAE: Mean Absolute Error.

**Table 3 ijerph-19-15638-t003:** Meteorological factors associated with monkeypox cases using ARIMAX and NB model.

Variables	ARIMAX Model			Negative Binomial Model		
	Coef.	95%CI	*p*-Value	Coef.	95%CI	*p*-Value
**Temperature**	51.56	−274.55 to 377.68	0.76	0.86	−0.42 to 2.13	0.13
**Dew/frost point**	−69.59	−366.61 to 227.43	0.65	−0.61	−1.93 to 0.71	0.19
**Relative humidity**	17.32	−83.71 to 118.35	0.74	−0.14	−0.53 to 0.26	0.36
**Precipitation**	−19.41	−57.25 to 18.42	0.31	−0.05	−0.19 to 0.08	0.51
**Surface pressure**	23.42	−9.9 to 56.75	0.17	−1.45	−3.29 to 0.4	0.45
**Wind**	−10.87	−45.67 to 23.92	0.54	−0.36	−1.82 to 1.09	0.12

## Data Availability

Not applicable.

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
