# Peer review of "Association between Global Monkeypox Cases and Meteorological Factors"

_ijerph, 2022, doi:10.3390/ijerph192315638_

Round 1
Reviewer 1 Report
The study investigates the potential relationship of climate change on one hand and the gobal outbreak of monkeypox on the other.
There are several minor and major concerns.
lines 30-32: "Determination" of various climate factors is not beneficial to "control", but to "understand" the outbreak.
line 41: "due point" should read "dew point", again in lines 52 and 349.
line 43: Dew point seems to have a negative, not a positive, impact.
Line 42: The authors state a "significant" impact, but Table 3 shows that there is no statistical significance. This "non-significance" is in agreement with the 95% CI, which includes zero.
In the abstract, it should be stated whether the statistical analysis refers to countries, regions, or continents.
Such a statement must also be included in the Materials and Methods section. I assume that it was a country-by-country analysis. Though this might well apply to the daily number of infections per million, there should also be an explanation how the metereological factors were defined across whole countries, which may even extend across different climatic zones.
Figure 1 states that there is no specific vaccine against monkeypox. Though smallpox vaccination is considered to have a protective effect against monkeypox, there is also a monkeypox vaccine already available.
lines 194-195: It is meaningless to compare the "variance" of different metereological factors, because they refer to completely different physical units.
Line 227: Separman rank correlation test, used in the Results section, must also be mentioned in the Material and Methods section.
In Figure 3, under the heading of "daily confirmed cases per country" and a legend referring to September 27, the US is labelled with >25.000 cases. When I checked "our world in data", there were 565 cases WORLDWIDE on this particular date. So I really cannot understand the meaning of the Figure. The magnitude of the numbers is about that of the CUMULATIVE incidence, but certainly not of the DAILY incidence.
"meteorological" is often erroneously replaced by "metrological".
Author Response
RESPONSE TO REVIEWER COMMENTS
The authors have addressed the reviewer's comments point by point. The resubmitted manuscript is used to present a detailed, point-by-point response addressing the specific comments and the reviewers’ concerns, including abstract, introduction, figures, tables, discussions, conclusions, and graphical abstract. All changes in the text are marked in track changed mode.
I do hope that all the issues and concerns raised by the reviewers have been addressed and that the revised manuscript will meet the standard IJERPH.

Reviewer 2 Report
The authors present a very interesting report on changes in MPXV cases over time and meteorological characteristics.
Although there may be some limitations to this approach (indeed the outbreak occurred during the transition from spring to summer), I believe that there may be more than some value to it.
The analyses are convincing and clearly presented. The manuscript is well written.
I would only suggest adding some references on the clinical manifestations (as well as atypical presentations/dermoscopy) to provide some further background:
- Maronese CA, Beretta A, Avallone G, et al. Clinical, dermoscopic and histopathological findings in localized human monkeypox: a case from northern Italy [published online ahead of print, 2022 Jul 13]. Br J Dermatol. 2022;10.1111/bjd.21773. doi:10.1111/bjd.21773
- Aromolo IF, Maronese CA, Avallone G, et al. Clinical spectrum of human monkeypox: An Italian single-centre case series [published online ahead of print, 2022 Sep 27]. J Eur Acad Dermatol Venereol. 2022;10.1111/jdv.18612. doi:10.1111/jdv.18612
- Maronese CA, Errichetti E, Avallone G, Beretta A, Marzano AV. Dermoscopy as a supportive diagnostic tool in human monkeypox [published online ahead of print, 2022 Sep 24]. J Eur Acad Dermatol Venereol. 2022;10.1111/jdv.18597. doi:10.1111/jdv.18597
Other than that I think that this is an outstanding piece of research. Great job!
Author Response

(The authors gave the same response as above.)

Reviewer 3 Report
The authors analyzed the influence of weather parameters on the spread of monkey pox. From the point of view of epidemiology and public health, such studies are important, but I have a few major comments about the research methodology:
1. The research covered a really short period of time
2. Data on the daily number of new cases come from 3 sources, of which only one can be considered reliable from the scientific point of view (WHO)
3. Cases of monkey pox are reported mainly in countries in the northern hemisphere (and Brazil in the tropical climate zone). The research period does not include winter in these countries.
4. Some factors, such as wind speed, dew point, may not necessarily influence human behavior as the monkey pox virus is transmitted by direct contact. Of course, there are still the factors of survival of the virus in the environment, which the period of summer with high humidity may be conducive to, however, in order to draw such conclusions, you need strong evidence covering several years of research.
Other comments are included in the pdf file

Author Response

(The authors gave the same response as above.)

Round 2
Reviewer 1 Report
The authors still do not explain the open question how they dealt with climate factors across large countries, which may definitely show different climatic features in different regions and subregions.
Author Response
The authors are thankful to the reviewer for his/her comments and added description in the method part as well as in the limitation of this study in the revised submitted version.
Furthermore, this study has another drawback for sorting climate factors, as capital cities are counted in this study rather than all cities in a country for lack of abundant, available online, and valid data sets.

Reviewer 3 Report
The authors revised manuscript including rev suggestions. Ms can be considered for publication.
Author Response
The authors are thankful to the reviewer for his/her comments for improving this manuscript quality.